# clusterBMA: Bayesian model averaging for clustering

**Owen Forbes**[1]*, **Edgar Santos-Fernandez**[1], **Paul Pao-Yen Wu**[1], **Hong-Bo Xie**[1,2], **Paul E. Schwenn**[3], **Jim Lagopoulos**[4], **Lia Mills**[4], **Dashiell D. Sacks**[4], **Daniel F. Hermens**[4], **Kerrie Mengersen**[1]

**1** Centre for Data Science, School of Mathematical Sciences, Queensland University of Technology, Brisbane, QLD, Australia, **2** School of Information Science and Engineering, Yunnan University, Kunming, China, **3** UQ Poche Centre for Indigenous Health, The University of Queensland, Brisbane, QLD, Australia, **4** Thompson Institute, University of the Sunshine Coast, Birtinya, QLD, Australia

* owen.forbes@hdr.qut.edu.au

**Data Availability Statement:** The conditions of the ethics approval do not permit public archiving of anonymised study data for the case study. Datasets from the Longitudinal Adolescent Brain Study are available on request from the data

## Abstract

Various methods have been developed to combine inference across multiple sets of results for unsupervised clustering, within the ensemble clustering literature. The approach of reporting results from one 'best' model out of several candidate clustering models generally ignores the uncertainty that arises from model selection, and results in inferences that are sensitive to the particular model and parameters chosen. Bayesian model averaging (BMA) is a popular approach for combining results across multiple models that offers some attractive benefits in this setting, including probabilistic interpretation of the combined cluster structure and quantification of model-based uncertainty. In this work we introduce *clusterBMA*, a method that enables weighted model averaging across results from multiple unsupervised clustering algorithms. We use clustering internal validation criteria to develop an approximation of the posterior model probability, used for weighting the results from each model. From a combined posterior similarity matrix representing a weighted average of the clustering solutions across models, we apply symmetric simplex matrix factorisation to calculate final probabilistic cluster allocations. In addition to outperforming other ensemble clustering methods on simulated data, *clusterBMA* offers unique features including probabilistic allocation to averaged clusters, combining allocation probabilities from 'hard' and 'soft' clustering algorithms, and measuring model-based uncertainty in averaged cluster allocation. This method is implemented in an accompanying R package of the same name. We use simulated datasets to explore the ability of the proposed technique to identify robust integrated clusters with varying levels of separation between subgroups, and with varying numbers of clusters between models. Benchmarking accuracy against four other ensemble methods previously demonstrated to be highly effective in the literature, *clusterBMA* matches or exceeds the performance of competing approaches under various conditions of dimensionality and cluster separation. *clusterBMA* substantially outperformed other ensemble methods for high dimensional simulated data with low cluster separation, with 1.16 to 7.12 times better performance as measured by the Adjusted Rand Index. We also explore the performance of this approach through a case study that aims to identify probabilistic clusters of individuals based on electroencephalography (EEG) data. In applied settings for

custodians at the Thompson Institute. Access will be granted to named individuals in accordance with ethical procedures governing the reuse of clinical data, including completion of a formal data sharing agreement. Contact LABSscmnti(at)usc.edu.au - more details at https://www.usc.edu.au/thompson-institute/research-at-the-thompson-institute/youth-mental-health/longitudinal-adolescent-brain-study/contact-labs. Code Availability All code written in support of this publication is publicly available at https://github.com/of2/clusterBMA, https://github.com/of2/cBMA_paper_simulations, and https://github.com/of2/EEG_clustering_public.

**Funding:** Full list of funders: Australian Research Council Centre of Excellence for Mathematical and Statistical Frontiers, CE140100049, KM Statistical Society of Australia, PhD Scholarship, OF Queensland University of Technology, PhD Scholarship, OF International Biometrics Society, PhD Scholarship, OF Prioritising Mental Health Initiative, Australian Commonwealth Government, DH.

**Competing interests:** The authors have declared that no competing interests exist.

clustering individuals based on health data, the features of probabilistic allocation and measurement of model-based uncertainty in averaged clusters are useful for clinical relevance and statistical communication.

## 1. Introduction

When faced with an unsupervised clustering problem, different clustering algorithms will often offer plausible, but different, perspectives on the clustering structure for a given dataset. A typical approach is to report results from one 'best' model based on goodness of fit, explainability, model parsimony, or other criteria. However, this ignores the uncertainty that arises from model selection, and results in inferences that are sensitive to the particular model and parameters chosen, and assumptions made, especially with small sample size data or when one or more of the clusters are relatively small [1]. Consideration of model-based uncertainty is particularly important when developing analyses with an 'M-open' perspective where the model class is not determined in advance, but is chosen and defined iteratively as more information becomes available and exploratory analysis proceeds [2, 3].

The problem of combining multiple sets of clustering results has received substantial attention in statistics and machine learning research, particularly in the ensemble clustering literature [4]. A common approach is to find something analogous to the 'median partition' between a number of clustering solutions [5]. An alternative, proposed in this paper, is to consider an approach based on Bayesian model averaging (BMA). BMA provides a framework that enables the analyst to probabilistically combine results across multiple models, where the contribution of each candidate model is weighted by its posterior model probability, given the data [3, 6]. Implementing BMA for clustering could allow integrated inference across multiple different clustering algorithms. Compared to other available approaches for combining clustering results, this approach has the potential to offer unique benefits including weighted averaging across models, generation of probabilistic inferences, incorporating the goodness of fit of the candidate algorithms, and quantification of model-based uncertainty. This would also enable downstream inferences based on combined clustering results to be calibrated for model-based uncertainty.

For clustering and other applications, BMA has typically been applied in the context of averaging within one class or family of models, using different combinations of potential explanatory variables [7]. Previous work in the space of BMA for clustering has been limited, with a few examples of applications within specific classes of clustering method such as Naive Bayes Classifiers and Gaussian Mixture Models [7, 8]. A gap remains in the literature around applying BMA across results from multiple different clustering algorithms. Ultimately all clustering methods predict quantities that can be compared directly across algorithms in a BMA framework, such as the marginal probability of individuals being allocated pairwise into the same cluster across all clusters. In previous work by Russell et al. on BMA for clustering with mixture models, the pairwise similarity matrix has been used to represent clustering results in a way that is directly comparable across sets of results regardless of the number and labels of clusters [8]. In this work the authors rely on the Bayesian Information Criterion (BIC) for each model, which they use to generate an approximation for the posterior model probability to assign weights for averaging results across models [9]. However, for the present application the BIC will not be directly comparable across results from different clustering algorithms, and for some algorithms it cannot be calculated at all where a likelihood term is not used.

The aim of this work is to propose a BMA framework that can effectively combine results across multiple unsupervised clustering algorithms. We showcase the performance and

effectiveness of this framework through various clustering applications, including simulated data experiments and a real-world case study involving neuroscientific data. Our method *clusterBMA* is designed to accommodate input solutions from a variety of clustering methods, and so we must look beyond the BIC to other measures to approximate the posterior model probability for each algorithm and enable weighted averaging. The BIC is commonly used as a clustering internal validation index (CIVI) for applications including choosing among candidate models with differing numbers of clusters. We propose considering alternative CIVIs, which share mathematical and conceptual similarity with the BIC, to approximate the marginal likelihood and generate an approximation for posterior model probability that is directly comparable across different clustering algorithms. *clusterBMA* shares useful features with some other ensemble clustering approaches, including being agnostic to the clustering algorithms used and the number of clusters in each individual set of results. Compared to other ensemble clustering methods, *clusterBMA* offers several unique and valuable features, including: probabilistic allocation to clusters averaged over multiple input models; combining allocation probabilities from 'hard' and 'soft' clustering algorithms; and measuring model-based uncertainty in averaged cluster allocation, which can be propagated forward for cluster-based inferences in a Bayesian setting to take that uncertainty into account. To our knowledge, no other ensemble clustering method has all of these key strengths.

In Section 2 we provide an overview of the methodological pipeline for *clusterBMA*, provide background on Bayesian model averaging in the context of clustering, and discuss approximation of posterior model probability based on clustering internal validation indices. In Section 3 we present methods and results for 3 simulations studies which include benchmarking *clusterBMA* against four other methods for ensemble clustering with simulated data, demonstrating handling of model-based uncertainty in relation to cluster separation, and performing model averaging across input solutions with different numbers of clusters. In Section 4 we present a case study applying our method for clustering electroencephalography data in young people, and highlight the utility of probabilistic allocations with quantified model-based uncertainty in a health research setting. In Section 5 we discuss the benefits and implications of this method and our findings, and consider limitations and future directions for this work.

## 2. Methods

In this section we present background, motivation and details of the methodological steps involved in *clusterBMA*. Table 1 presents a comparison of features that are available in *clusterBMA*, and five other available methods for ensemble clustering. The features of BMA for Gaussian Mixture Models are presented for the sake of comparison to previous work which has developed a BMA approach within one class of clustering algorithm [8]. We selected the other four ensemble methods as they have been shown in the literature to be effective, and they are readily compared with *clusterBMA* through their implementation in the *diceR* R package [4, 10]. Relative to these other methods, *clusterBMA* uniquely offers several features including combining cluster allocation probabilities across 'soft' and 'hard' clustering algorithms, generating probabilistic allocations to averaged final clusters, and quantifying model-based uncertainty. These features are discussed in more detail throughout the following sections.

### 2.1 Bayesian model averaging for clustering

Consider a quantity of interest $\Delta$ which is present in every model across a set of candidate models for a given analysis. Given data $Y$ with dimension $D$, consider a set of posterior estimates $\Delta_m$, $m = 1, \ldots, M$, each obtained from a corresponding model $\mathcal{M}_m$. The BMA

**Table 1. Feature comparison between *clusterBMA* and five other ensemble clustering methods.**

| Method | Combine solutions with different numbers of clusters | Combine solutions from different algorithms | Weight each input solution by model quality | Combine allocation probabilities from 'soft' and 'hard' clustering algorithms [1] | Probabilistic allocation to averaged clusters | Measure model-based uncertainty in allocations to averaged clusters |
|---|---|---|---|---|---|---|
| BMA for GMM [2] | ✓ | ✗ | ✓ | ✗ | ✗ | ✗ |
| CSPA | ✓ | ✓ | * | ✗ | ✗ | ✗ |
| LCE | ✓ | ✓ | * | ✗ | ✗ | ✗ |
| K modes | ✓ | ✓ | * | ✗ | ✗ | ✗ |
| Majority voting | ✓ | ✓ | * | ✗ | ✗ | ✗ |
| *clusterBMA* | ✓ | ✓ | ✓ | ✓ | ✓ | ✓ |

[1] 'Soft' clustering refers to algorithms which assign a probability between 0 and 1 for each observation to be allocated to each cluster (e.g. Gaussian Mixture Model).

'Hard' clustering refers to algorithms which assign a 0 or 1 binary probability of cluster allocation (e.g. $k$-means).

[2] Features of BMA for Gaussian Mixture Models are based on the preprint by Russell et al [8].

* For these ensemble methods, weighting by internal validation indices is available as an auxiliary feature in the *diceR* R package [10].

CSPA = Cluster-based Similarity Partioning Algorithm; LCE = Linkage-Based Cluster Ensembles.

framework provides a weighted average of these estimates, given by

$$p(\Delta|Y) = \sum_{m=1}^{M} p(\Delta_m|Y, \mathcal{M}_m)p(\mathcal{M}_m|Y), \qquad (1)$$

where $p(\mathcal{M}_m|Y)$ is the posterior probability of model $\mathcal{M}_m$, given by

$$p(\mathcal{M}_m|Y) = \frac{p(Y|\mathcal{M}_m)p(\mathcal{M}_m)}{\sum_{m'=1}^{M} p(Y|\mathcal{M}_{m'})p(\mathcal{M}_{m'})}. \qquad (2)$$

Here $p(\mathcal{M}_m)$ is the prior probability for each model, and $p(Y|\mathcal{M}_m)$ is the marginal likelihood for each model (also called the model evidence) [7]. As is common in BMA applications, here we assign priors to give equal weight to each model, with $p(\mathcal{M}_m) = \frac{1}{M}$. Alternative approaches could be used for assigning prior probability for each model, as addressed in the Discussion.

Following Russell et al. [8], we consider the pairwise similarity matrix as a common property $\Delta$ which is present across all clustering algorithms of interest. The similarity matrix $S^m$ of pairwise co-assignment probabilities for any clustering model $\mathcal{M}_m$ will have dimensions $N \times N$. Since clustering solutions are combined at the level of pairwise co-allocation probabilities via similarity matrices, this has the benefit of avoiding any issues regarding alignment of cluster labels across the different models. Each element $s_{ij}$ of the similarity matrix represents the probability that data points $i$ and $j$ belong to the same cluster $g_k \forall k = 1, \ldots, K_m$ where $K_m$ is the total number of clusters in model $\mathcal{M}_m$:

$$s_{ij}|\mathcal{M}_m, Y = \sum_{k=1}^{K_m} p(g_k|i, \mathcal{M}_m) \, p(g_k|j, \mathcal{M}_m), \qquad (3)$$

where $g_k$ is the $k$th cluster, and $p(g_k|i, \mathcal{M}_m)$ indicates the probability that point $i$ is a member of cluster $g_k$ in model $\mathcal{M}_m$ [11]. Here $\Delta_m = S^m = \{s_{ij}\}$, $i, j = 1, \ldots, N$. For 'hard' clustering methods such as k-means or agglomerative hierarchical clustering, these pairwise probabilities will be 0 or 1, while for 'soft' clustering methods such as a Gaussian mixture model, these pairwise probabilities can take any value between 0 and 1. To represent each clustering solution as a pairwise

similarity matrix $S^m$, we can use the $N \times K_m$ matrix $A^m$ of cluster allocation probabilities where each element $A_{ik}^m$ contains the probability that point $i$ is allocated to cluster $k$ under model $\mathcal{M}_m$. We calculate the similarity matrix for each model by multiplying $A^m$ by its transpose, and setting the diagonal to 1 [8]:

$$S_{ij}^m = \begin{cases} (A^m(A^m)^\top)_{ij} & \text{if } i \neq j \\ 1 & \text{if } i = j \end{cases}. \tag{4}$$

## 2.2 Approximating posterior model probability with clustering internal validation indices

As introduced above, BMA has previously been implemented within the model class of Gaussian mixture models, by calculating an element-wise weighted average across the similarity matrices representing each set of clustering results [8]. These authors weighted results from each mixture model according to an approximation of posterior model probability based on the BIC. Assuming equal prior probability for each candidate model, this is equivalent to weighting each model by its adjusted marginal likelihood as a proportion of the sum of the adjusted marginal likelihoods across all candidate models:

$$P(\mathcal{M}_m|Y) \approx \frac{exp\left(\frac{1}{2}BIC_m\right)}{\sum_{m=1}^M exp\left(\frac{1}{2}BIC_m\right)}, \tag{5}$$

where

$$BIC_m = 2\log(\mathcal{L}) - \kappa_m \log(N). \tag{6}$$

Here $\mathcal{L}$ is the likelihood of the data given the model, $\kappa_m$ is the number of estimated model parameters for the model, and $N$ is the number of observations [12]. This is the negative of the usual construction of the BIC, and a larger number of model parameters $\kappa_m$ will result in a smaller estimate for the approximated posterior model probability of model $\mathcal{M}_m$. The BIC has a theoretically established use for estimating marginal likelihood and posterior model probability in the context of Gaussian mixture models [9]. It can be seen from Eqs 5 and 6 that the weighting method used by Russell et al. is constructed to recover an estimate for the likelihood [8]. From Eq 6, the likelihood $\mathcal{L}$ is assumed to be a multivariate Gaussian mixture, calculated as:

$$\mathcal{L}(\Theta) = \sum_{n=1}^N \sum_{k=1}^K \pi_k \mathcal{N}(\mathbf{y}_n|\boldsymbol{\mu}_k, \boldsymbol{\Sigma}_k), \tag{7}$$

where $\boldsymbol{\mu}$ is a $D$-dimensional vector of means, $\pi_1, \ldots, \pi_K$ are the mixing probabilities used to weight each component distribution, $K$ is the selected number of mixture components, $\Sigma$ is a $D \times D$ covariance matrix, $|\Sigma|$ represents the determinant of $\Sigma$, and $\mathcal{N}(\mathbf{x}_n|\boldsymbol{\mu}_k, \boldsymbol{\Sigma}_k)$ is a multivariate Gaussian density given by

$$\mathcal{N}(\mathbf{y}|\boldsymbol{\mu}, \boldsymbol{\Sigma}) = \frac{exp\{-\frac{1}{2}(\mathbf{y}-\boldsymbol{\mu})^T \boldsymbol{\Sigma}^{-1}(\mathbf{y}-\boldsymbol{\mu})\}}{|\Sigma|^{\frac{1}{2}}(2\pi)^{\frac{D}{2}}}. \tag{8}$$

While ideally we would like to use a measure such as the BIC with strong theoretical support for approximating the marginal likelihood to weight each model, the BIC is not viable for our application of weighting solutions generated from multiple classes of clustering algorithm.

The BIC is theoretically supported for estimating the posterior model probability for GMMs, but in practice the BIC has a number of shortcomings for the purpose of estimating posterior model probability in the context of Bayesian model averaging for clustering solutions generated by multiple algorithms. For example, three considerations are as follows. First, BIC scores are not able to be directly compared across multiple classes of clustering algorithm, and are not able to be generated at all for some classes of clustering algorithm without a likelihood term, such as hierarchical clustering. Second, the exponentiation step in Eq 5 required to estimate the marginal likelihood from the BIC tends to result in a large majority of the overall weight being assigned to one model. Some evidence suggests that in this way the BIC works well for model selection (assigning all weight to a single model), but not as well for model averaging [13, 14]. Third, there are known instances where the BIC is not a good reflection of clustering analytic objectives. For example, the BIC has well documented difficulties for model selection in high dimensional settings [15], including a tendency towards underfitting and selecting overly parsimonious mixture models with too few mixture components [16, 17].

Instead of the BIC, we can consider cluster internal validation indices as a set of measures which offer methods for assessing model quality and approximating posterior model probability across clustering algorithms with different constructions and objective functions [18]. CIVIs are typically developed to reflect common traits of clustering analytic objectives shared across algorithms including compactness, separation or inter-cluster density for a particular clustering of a dataset. Compactness describes how closely related the data points are within a cluster, and is typically measured by within-cluster variance or sum of squared distances of all points from their respective cluster centres. Separation describes how distinct clusters are from each other, and is often measured by the distances between cluster centres or minimum pairwise distances between points across clusters [19]. Similar to cluster separation, the goal for inter-cluster density is that the density of points in the area between clusters is low in comparison with the density within the considered clusters [20]. The BIC can be applied as a CIVI, for purposes including choosing a suitable number of clusters for a finite mixture model. From Eqs 6–8 it can be seen that in the context of Gaussian mixture models, the BIC is driven by a ratio of within-cluster variance (compactness) to overall variance.

Internal validation indices are commonly interpreted in a way that is analogous to the marginal likelihood, being used to make some judgement about model quality or goodness of fit in order to decide between multiple candidate models with differing numbers of cluster $K_m$. There are established parallels between different CIVIs and objective functions, loss functions and likelihoods for clustering algorithms. Some CIVIs have similar structures to the objective functions of algorithms for which they were developed and will tend to preference results generated by those algorithms [21]. For instance, the Xie-Beni index has a clear link to the objective function for the Fuzzy C-Means algorithm [22]. Other indices are developed to reflect more general analytical objectives that are common across the likelihoods or objective functions for many clustering algorithms, such as the Calinski-Harabasz index [23], or the S_Dbw [20], among others [18]. CIVIs have been used as loss functions for clustering with neural networks [24], and for measuring and comparing model quality across different clustering algorithms [25, 26].

While it would be preferable to start from an estimation of the marginal likelihood for each model to approximate posterior model probability, in this application this is not viable. Instead we take the approach of starting from an internal validation index to substitute for the marginal likelihood term, and building an approximation for the posterior model probability to weight each candidate clustering solution.

We acknowledge that the process of selecting an appropriate CIVI for model weighting opens the door to myriad candidate validation indices, from which the analyst must choose an

appropriate measure to suit the goals of their clustering analysis. We view this as a strength and a designed feature of our method as it does not automate the choice of an appropriate measure to weight models across all scenarios, and instead requires the analyst to consider and choose a weighting measure that is appropriate for the clustering analytic objectives of their application. Just as the analyst must make reasoned and considered decisions about preparing data, choosing appropriate clustering algorithms, and choosing appropriate numbers of clusters, a CIVI should be chosen that is appropriate for weighting each model in a given analysis. In this paper we make recommendations regarding two indices which are likely to perform well for approximating clustering posterior model probability in many common applications —the Calinski-Harabasz (CH) and S_Dbw indices [20, 23]. These two indices are well supported with evidence regarding their utility for comparing model quality across solutions generated from different clustering algorithms [25], and their robustness to different challenging features of clustering data [27]. Both of these were developed as algorithm-independent indices, reflecting general clustering analytic objectives such as cluster compactness, cluster separation, and inter-cluster density, and reducing bias towards any one class of algorithm. We address some caveats and limitations of these indices in the Discussion.

Similarly to the BIC which is driven by a ratio of cluster compactness to overall variance for GMMs, the CH index is an internal validity measure representing a ratio of cluster separation to compactness, calculated with a ratio of between-cluster sums of squares to within-cluster sums of squares, penalised by the number of clusters in the model [23]. This index is calculated as:

$$CH = \frac{\sum_{j \neq k}^{K} n_k\, d^2(c_j, c_k)/(K-1)}{\sum_{k=1}^{K} \sum_{x \in g_k} d^2(x, c_k)/(N-K)},$$ (9)

where $n_k$ is the number of observations allocated to cluster $g_k$, $d(x, y)$ is the distance between $x$ and $y$, $c_k$ is the centroid of $g_k$, and $x \in g_k$ are the data points allocated to cluster $g_k$. Higher CH scores indicate better internal clustering validity, with more separated and compact clusters.

The S_Dbw index is calculated as the sum of an intra-cluster variance term $Scat(K)$ that measures cluster compactness, and a density term $Dens\_bw(K)$ that measures inter-cluster density:

$$S\_Dbw = Scat(K) + Dens\_bw(K).$$ (10)

The intra-cluster variance term $Scat(K)$ is defined as:

$$Scat(K) = \frac{1}{K} \sum_{k=1}^{K} \frac{||\sigma(C_k)||}{||\sigma(D)||},$$ (11)

where $\sigma(C_k)$ is the variance of cluster $C_k$ and $\sigma(D)$ is the variance of the dataset. The inter-cluster density term $Dens\_bw(K)$ is defined as:

$$Dens\_bw(K) \quad = \frac{1}{K(K-1)} \sum_{k=1}^{K} \left( \sum_{j \neq k}^{K} \frac{\sum_{x \in C_k \cup C_j} f(x, u_{kj})}{max\left( \sum_{x \in C_k} f(x, c_k), \sum_{x \in C_j} f(x, c_j) \right)} \right)$$

$$f(x, y) \quad = \begin{cases} 0 & \text{if } d(x, y) > \frac{1}{K}\sqrt{\sum_{k=1}^{K} ||\sigma(C_k)||} \\ 1 & \text{otherwise,} \end{cases}$$ (12)

where $u_{kj}$ is the mid-point between $c_k$ and $c_j$. $Dens\_bw$ represents a ratio of inter-cluster density

to within cluster density, with lower values indicating better separation between clusters. Lower S_Dbw scores indicate better internal clustering validity, with more compact and well-separated clusters.

Having chosen a CIVI to act as a weighting variable $\mathcal{W}_m$ for each model, we propose the following normalised weight $\hat{\mathcal{W}}_m$ as an approximation for the marginal likelihood $P(Y|\mathcal{M}_m)$ for each model:

$$P(Y|\mathcal{M}_m) \approx \hat{\mathcal{W}}_m \coloneqq \begin{cases} \dfrac{\mathcal{W}_m}{\sum_{m'=1}^{M} \mathcal{W}_{m'}} & \text{if } \mathcal{W}_m \text{ is to be maximised} \\[4ex] \dfrac{\frac{1}{\mathcal{W}_m}}{\sum_{m'=1}^{M} \frac{1}{\mathcal{W}_{m'}}} & \text{if } \mathcal{W}_m \text{ is to be minimised,} \end{cases} \qquad (13)$$

where $\coloneqq$ indicates 'is defined as'. Using this approximation of the marginal likelihood in Eq 13 and setting equal prior probability for all input models $p(\mathcal{M}_m) = \frac{1}{M}$, we arrive at the following approximation for posterior model probability, substituting in to Eq 2:

$$P(\mathcal{M}_m|Y) \approx \frac{\hat{\mathcal{W}}_m\left(\frac{1}{M}\right)}{\sum_{m'=1}^{M} \hat{\mathcal{W}}_{m'}\left(\frac{1}{M}\right)} = \hat{\mathcal{W}}_m. \qquad (14)$$

While these are the two indices that we recommend due to evidence of their strong performance across a range of algorithms and settings, users of the *clusterBMA* package can select any cluster internal validation index implemented in the *clusterCrit* R package. Details on available cluster internal validation indices and their interpretation (e.g. whether to be maximised or minimised) are provided in a previous publication, and in documentation accompanying the package [28].

## 2.3 Symmetric simplex matrix factorisation for probabilistic cluster allocation

Having represented each candidate set of clustering results as a similarity matrix as in Eq 4 and having calculated normalised weights as in Eq 13, we can generate a consensus matrix $C$ which is a posterior similarity matrix of co-assignment probabilities, using a weighted average of similarity matrices from input models $S^m$, $m = 1, \ldots, M$. We calculate the $N \times N$ consensus matrix $C$ as the element-wise weighted average of the similarity matrices from each candidate model, weighted by the normalised weights $\hat{\mathcal{W}}_m$:

$$C = \sum_{m=1}^{M} \hat{\mathcal{W}}_m S_m. \qquad (15)$$

We then generate final probabilistic cluster allocations based on this consensus matrix using symmetric simplex matrix factorisation (SSMF), a method developed in the context of an approximate Bayesian method for clustering [29], and applied for Bayesian distance clustering [30]. Having specified a final number of clusters $K_{BMA}$, this method can be used to factorise an $N \times N$ posterior similarity matrix, in this case the consensus matrix $C$, into an $N \times K_{BMA}$ matrix of cluster allocation probabilities resulting from this BMA pipeline, $A^{BMA}$.

For each input model, we suggest choosing the optimal number of clusters $K_m$ based on a variety of cluster internal validation indices. To select the number of clusters for the final BMA clustering solution $K_{BMA}$, one possible heuristic is to select the largest $K_m$ across the input

models. SSMF as implemented by Duan includes an L2 regularisation step [29], which is useful for emptying redundant clusters in the final clustering results represented in $A^{BMA}$. This L2 regularisation step can result in fewer final clusters than selected for $K_{BMA}$. For instance, where a model $\mathcal{M}_f$ with fewer clusters $K_f$ is heavily weighted by $\hat{\mathcal{W}}_f$ relative to other models, the *clusterBMA* solution may contain $K_f$ combined clusters after L2 regularisation even where a larger $K_{BMA}$ is selected. Given the reduction of redundant clusters with L2 regularisation, another possible heuristic for choosing $K_{BMA}$ would be to choose a larger number of clusters than the largest $K_m$ across the input models, accommodating the possibility of different sets of sub-clusters appearing across different input models.

This method enables quantification of uncertainty for probabilistic cluster allocations. Following previous work in Bayesian clustering, we can measure uncertainty in this allocation as the probability that the estimated cluster allocation $g_i$ for point $i$ is not equal to the 'true' cluster allocation $\hat{g}_i$ for that point, $p(g_i \neq \hat{g}_i)$ [30]. This uncertainty measure incorporates both within-model and across-model uncertainty for cluster allocation from the input candidate models.

### 2.4 *clusterBMA* overview

Here we present a high level overview of the methodological steps involved in *clusterBMA*. The intention is to provide a reference structure for the reader, making the detailed explanations of each individual step easier to understand in the broader context of this framework. The method implemented in *clusterBMA* consists of the following steps:

1. Calculate results from multiple clustering algorithms on the same dataset. These clustering solutions can be produced by any 'hard' (binary allocation, e.g. *k*-means or hierarchical clustering) or 'soft' (probabilistic allocation, e.g. Gaussian mixture model) clustering algorithm [19], and can each contain a varying number of clusters. Results from each model should be in the form of a $N \times K_m$ allocation matrix $A^m$, where $N$ is the number of data points, $k = 1, \ldots, K$ indexes the clusters in the model, and $m = 1, \ldots, M$ indexes the input models.

2. Represent the clustering solution $A^m$ as a $N \times N$ pairwise similarity matrix $S^m$.

3. Compute an approximate posterior model probability $\hat{\mathcal{W}}_m$ to weight each input solution, calculated as a normalised weight based on a CIVI such as the CH or S_Dbw indices [20, 23].

4. Calculate the consensus matrix $C$ as an element-wise weighted average across the similarity matrices $S^m$, $m = 1, \ldots, M$ from (2), weighted by the approximation for posterior model probability from (3).

5. Generate a final set of averaged probabilistic cluster allocations using symmetric simplex matrix factorisation of the consensus matrix in (4), a method proposed in an approximate Bayesian clustering context [29].

An R package *clusterBMA* implementing this method has been developed and made available on Github [31].

## 3. Simulation studies

To investigate the performance and properties of *clusterBMA*, we conducted three simulation studies. The first simulation study aimed to benchmark *clusterBMA* against several other ensemble clustering methods that have been shown in the literature to be effective [4, 10], assessing their performance under conditions of varying numbers of dimensions and

levels of cluster separation for simulated data. The aim of the second simulation study was to investigate the effect of cluster separation on model averaging results, and to test the utility of *clusterBMA* for identifying model-based uncertainty in situations of increasing ambiguity between clustering solutions. The objective of the third simulation study was to demonstrate the ability of *clusterBMA* to average across models with differing numbers of clusters. Full details of methods and results for simulation studies 2 and 3 are presented in the S1 File.

### 3.1 Simulation study 1–methods

We designed this principal simulation study to compare the performance of *clusterBMA* with several other ensemble clustering algorithms, using simulated datasets with low (2), medium (10) and high (50) numbers of dimensions, and varying conditions of low, medium and high levels of separation between simulated clusters.

We generated 10 replicates of simulated datasets under 9 combinations of each level of cluster separation and number of dimensions. Simulated datasets were generated using the R package *clusterGenerate*, resulting in a total of 90 simulated datasets [32]. This package allows the user to simulate data from multivariate normal clusters, and easily control the degree of separation between the clusters. Each simulated dataset contained 1500 data points, with 500 points in each of 3 clusters. For each number of dimensions (2, 10 and 50) we generated 10 high separation datasets (separation value = 0.1), 10 medium separation datasets (separation value = -0.05), and 10 low separation datasets (separation value = -0.15). These separation values were chosen heuristically through trial and error, based on visual inspection of the plotted values in 2 dimensions.

For each simulated dataset, we calculated clustering solutions with $k = 3$ clusters using 9 clustering algorithms: Hierarchical Clustering with average linkage, using the R base package *stats* [33, 34]; Divisive Analysis Clustering (DIANA), using the R package *cluster* [35, 36]; *k*-means, using the R base package *stats* [34, 37]; Partitioning Around Medoids (PAM), using the R package *cluster* [35, 36]; Affinity Propagation, using the R package *apcluster* [38, 39]; Spectral Clustering, using the R package *kernlab* [40, 41]; Gaussian Mixture Model (GMM), using the R package *mclust* [42, 43]; Self-Organising Maps (SOM), using the R package *kohonen* [44]; and Fuzzy C-Means, using the R package *e1071* [45, 46].

For each set of 9 clustering solutions, we combined results across these algorithms using *clusterBMA*, and compared its performance to four other cluster ensemble methods. We used the Calinski-Harabasz index as the CIVI for *clusterBMA* weighting in Eq 13, as the data were generated from multivariate normal clusters and clusters were approximately spherical. The other cluster ensemble methods included for comparison against *clusterBMA* were the Cluster-based Similarity Partitioning Algorithm (CSPA) [47], Linkage-Based Cluster Ensembles (LCE) [48], K-modes [49], and Majority Voting [50]. These ensemble methods were applied using the R package *diceR* [10]. Performance for each ensemble method was assessed using the Adjusted Rand Index (ARI) to assess the degree of agreement between the combined clustering solution from each ensemble method and the true cluster labels for the simulated data [51]. The ARI was calculated using the R package *pdfCluster* [52].

For each dataset, we also calculated the ARI for *clusterBMA* using a subset of the data points which had a high probability ($p > 0.8$) of allocation to the final averaged clusters for each dataset. This allowed us to demonstrate a unique feature of *clusterBMA* relative to these other methods, where it enables probabilistic allocation to final ensemble clusters and measures model-based uncertainty arising from ambiguity or disagreement between different clustering solutions. This feature can be used to confine cluster-based inference to be made only for those points which have low model-based uncertainty, and refrain from making clustering

inferences for points with a high degree of ambiguity in their cluster allocation across different input solutions.

## 3.2 Simulation study 1–results

Table 2 presents the mean and standard deviation for ARI scores between each ensemble solution and the true cluster labels, for 10 simulated datasets in each combination of cluster separation level (high, medium, low) and number of dimensions for the simulated dataset (2, 10, 50). Examples of 2-dimensional datasets at each level of cluster separation are visualised in Fig 1. S1 Table in S1 File presents the mean model weights assigned to each of the 9 clustering algorithms, across the combinations of separation levels and numbers of dimensions.

These results shows that *clusterBMA* had similar or better performance relative to the best performing alternative ensemble methods under all conditions, and substantially outperformed all competing ensemble methods for 50-dimensional simulated data with medium or low separation between clusters. For 50-dimensional simulated data with low cluster separation, performance measured by mean ARI for *clusterBMA* (ARI = 0.57) was 1.16 (CSPA ARI = 0.49) to 7.12 (Majority voting ARI = 0.08) times better than competing ensemble methods.

Further, when considering only points with high probability ($p > 0.8$) of allocation to final clusters, *clusterBMA* offered much higher accuracy across all datasets when confining inference to points with low model-based uncertainty in the model averaged solution. The proportion of points with ($p > 0.8$) of allocation to final clusters varied from 0.97 (High separation, 2 dimensions) to 0.67 (Low separation, 50 dimensions).

**Table 2. Simulation study results—ARI mean (standard deviation) across 10 simulated datasets, comparing clusterBMA with four other ensemble clustering methods.**

| Cluster separation | Method | 2 Dimensions | 10 Dimensions | 50 Dimensions |
|---|---|---|---|---|
| High | CSPA | 0.93 (0.03) | 0.95 (0.01) | 0.94 (0.02) |
| | LCE | 0.89 (0.13) | 0.91 (0.13) | 0.74 (0.22) |
| | K-modes | 0.93 (0.03) | 0.89 (0.18) | 0.93 (0.01) |
| | Majority voting | 0.93 (0.03) | 0.89 (0.15) | 0.38 (0.24) |
| | clusterBMA | 0.93 (0.03) | 0.95 (0.01) | 0.94 (0.02) |
| | clusterBMA—high certainty | 0.97 (0.01) | 0.98 (0.01) | 0.97 (0.01) |
| Medium | CSPA | 0.81 (0.05) | 0.79 (0.03) | 0.69 (0.16) |
| | LCE | 0.81 (0.05) | 0.76 (0.09) | 0.42 (0.25) |
| | K-modes | 0.81 (0.04) | 0.80 (0.02) | 0.68 (0.17) |
| | Majority voting | 0.81 (0.04) | 0.68 (0.16) | 0.11 (0.09) |
| | clusterBMA | 0.81 (0.04) | 0.80 (0.02) | 0.76 (0.02) |
| | clusterBMA—high certainty | 0.86 (0.04) | 0.91 (0.02) | 0.86 (0.04) |
| Low | CSPA | 0.63 (0.08) | 0.62 (0.05) | 0.49 (0.16) |
| | LCE | 0.60 (0.11) | 0.61 (0.05) | 0.32 (0.17) |
| | K-modes | 0.63 (0.07) | 0.58 (0.11) | 0.49 (0.18) |
| | Majority voting | 0.62 (0.09) | 0.42 (0.15) | 0.08 (0.10) |
| | clusterBMA | 0.63 (0.08) | 0.61 (0.04) | 0.57 (0.13) |
| | clusterBMA—high certainty | 0.70 (0.06) | 0.78 (0.04) | 0.69 (0.11) |

ARI was calculated for 10 datasets in each combination of three clustering separation levels (Low, Medium and High) and differing number of dimensions (2, 10 and 50).

ARI = Adjusted Rand Index; CSPA = Cluster-based Similarity Partioning Algorithm; LCE = Linkage Clustering Ensemble; "clusterBMA—high certainty" indicates ARI for clusterBMA based on points with allocation probability $p > 0.8$ to final clusters.

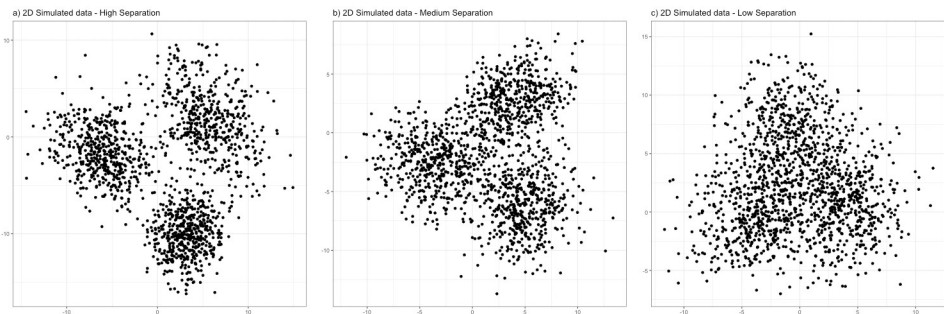

**Fig 1. Example scatter plots of 2-dimensional simulated datasets at each level of cluster separation.**

### 3.3 Simulation study 2

In the second simulation study we aimed to demonstrate the strength of *clusterBMA* for incorporating and accounting for the uncertainty that arises across multiple candidate models, when trying to identify cluster structure in data that may be representing overlapping or poorly separated groups. We generated three simulated datasets using the R package cluster-Generate. Each simulated dataset contained 500 data points, with 100 points in each of 5 clusters, at three levels of separation between clusters (S1 Fig in S1 File). For each simulated dataset we applied *k*-means, hierarchical clustering using Ward's method, and Gaussian mixture model, selecting the number of clusters $K_m = 5$ for each. We combined each set of three clustering solutions using *clusterBMA* with $K_{BMA}$ set to 5, and the Calinski-Harabasz index as the CIVI for weighting.

S5 Fig in S1 File presents the final cluster allocations for each simulated dataset generated by *clusterBMA*, with point size scaled according to uncertainty of cluster allocation, where larger points representing higher uncertainty of allocation to a final cluster. It is evident that as the degree of separation present between clusters in the data becomes lower, the degree of uncertainty in cluster allocations rises due to ambiguity and disagreement in clustering results across the multiple input algorithms. As real world data will typically not have clearly separated clusters, this demonstrates that in these common scenarios with messy data overlapping among possible clusters, it is valuable to use this Bayesian model averaging approach to take model-based uncertainty into account. Our method incorporates and quantifies this uncertainty, enabling cluster-based inferences that are better calibrated for this model-based source of uncertainty that is often ignored when using results from one chosen clustering algorithm.

### 3.4 Simulation study 3

The third simulation study demonstrates the ability of *clusterBMA* to average across models with differing numbers of clusters. We generated a simulated dataset contained 300 data points, with 100 points in each of 3 clusters. We calculated two clustering solutions: *k*-means with K = 3, and hierarchical clustering with K = 2. As above, we applied the *clusterBMA* pipeline to combine results from these two models, with $K_{BMA}$ set to 3.

S6 Fig in S1 File displays the clustering solutions generated by each algorithm, and the combined solution from *clusterBMA*. From panel (c) in S6 Fig in S1 File, there is low model-based uncertainty for cluster 1, moderate model-based uncertainty for clusters 2 and 3 where the algorithms disagree on the number of clusters for these points, and high model-based uncertainty at the border of cluster 2 with cluster 1 where the algorithms disagree on the allocation

of marginal points. These results demonstrate that our approach can combine clustering solutions across models with with differing numbers of clusters $K_m$.

# 4. Case study: Clustering adolescents based on resting state EEG recordings

We demonstrate the application of this method through a case study to identify clusters of adolescents based on resting state electroencephalography (EEG) recordings. Three popular unsupervised clustering algorithms were applied, and each provided a different perspective on the clustering structure in the data. To quantify model-based uncertainty and enable probabilistic inference about clustering structure by combining results across the candidate models, we implemented the *clusterBMA* framework described above. The full details of the data, pre-processing, dimension reduction and clustering analyses for this case study are presented in a previous publication [53].

## 4.1 Case study methods

In this section we present a case study applying *clusterBMA* in the scenario of clustering young people based on resting state electroencephalography data, and highlight the utility of probabilistic allocations with quantified model-based uncertainty in an applied health research setting.

**4.1.1 Data collection.**   Resting state, eyes-closed EEG data were collected as part of the Longitudinal Adolescent Brain Study (LABS) conducted at the Thompson Institute in Queensland, Australia. LABS is a longitudinal cohort study examining the interactions between environmental and psychosocial risk factors, and outcomes including cognition, self-report mental health symptoms, neuroimaging measures, and psychiatric diagnoses [54]. The present study uses data collected from (N = 59) participants at the first time point in the study, from 12-year-old participants (Mean = 12.64, SD = 0.32). Participants were recruited between July 2018 and June 2020. For data used in this paper, authors did not have access to information that could identify individual participants during or after data collection. Further information on data collection and study protocols for LABS are provided in previous publications [54, 55]. In this paper we aim to identify data-driven subgroups of LABS participants using EEG data.

**4.1.2 Ethical approval.**   LABS received ethical approval from the University of the Sunshine Coast Human Research Ethics Committee (Approval Number: A181064). Written informed assent and consent was obtained from all participants and their guardian/s. For data analysis conducted at the Queensland University of Technology (QUT), the QUT Human Research Ethics Committee assessed this research as meeting the conditions for exemption from HREC review and approval in accordance with section 5.1.22 of the National Statement on Ethical Conduct in Human Research (2007). Exemption Number: 2021000159.

## 4.2 Statistical analyses

This case study involved a multi-stage analysis pipeline. The first stage for clustering based on EEG frequency characteristics included automated EEG pre-processing [53], frequency decomposition with multitaper analysis [56, 57], and selection and calculation of 8 summary features in the frequency domain. The second stage included dimensionality reduction using principal component analysis, and applying three popular unsupervised clustering algorithms to this dimension-reduced data: *k*-means [37], hierarchical clustering using Ward's method [33], and a Gaussian Mixture Model (GMM) [42].

We calculated results for *k*-means using the kmeans() function with default settings from the base package 'stats' in R [34]. We calculated results for hierarchical clustering using the hclust() function with method 'ward.D2' from the base package 'stats' in R. We calculated

results for GMM with default settings including Euclidean distance and a diagonal covariance matrix using the R package 'ClusterR' [58]. For the calculation of internal validation indices for GMM results in this work, we use the crisp projection with allocation of each data point to the mixture component in which it has the highest allocation probability.

For each clustering algorithm, the optimal number of clusters $K_m$ was selected on the basis of a number of internal validation indices which could be calculated for all three methods. Internal validation indices can be used to identify the number of clusters that creates the most compact and well-separated set of subgroups in the data [19]. Each index is calculated based on a slightly different construction, so using a selection of multiple indices can be more robust than relying on a single index. Indices used included the Dunn index [59], silhouette coefficient [60], Davies-Bouldin index [61], Calinski-Harabasz index [23], and Xie-Beni index [22]. Clustering internal validation indices were calculated using the R package *clusterCrit* [28].

To probabilistically combine clustering results across these three algorithms, we implemented *clusterBMA* using the Calinski-Harabasz index to generate weights for each model, as all of the algorithms appeared to generate clusters with approximately spherical variance in 3 dimensions, and we did not have any *a priori* reasons to expect strongly non-spherical clusters. Each set of cluster allocations was represented as a similarity matrix, and a normalised weight $\hat{\mathcal{W}}_m$ was calculated using Eqs 11 and 12. Subsequently we calculated an element-wise weighted average across the similarity matrices using these weights, producing a consensus matrix. From this consensus matrix we applied symmetric simplex matrix factorisation to generate final probabilistic cluster allocations with associated uncertainty.

## 4.3 Case study results

From the principal component analysis, the first three principal components were retained which together explained 80.6% of the overall variance. On the basis of the internal validation criteria introduced above, we chose to implement a 5-cluster solution in each of the three individual clustering methods. Further details on selecting the number of clusters $K_m$ for each method are provided in a previous publication [53]. Table 2 presents the number of individuals assigned to each cluster for the three clustering algorithms, and to the final clusters generated from *clusterBMA*. For each algorithm, cluster labels (1–5) have been assigned by decreasing cluster size except for HC clusters 3 and 4, for which labels were switched for the sake of clearer visual comparison across plots in Fig 2. This relabeling step was applied only for the sake of visual clarity, as *clusterBMA* does not require cluster labels to be aligned across the candidate models. Fig 2 presents the clustering results from each algorithm, plotted according to each two-dimensional combination of the three retained principal components. This figure indicates that there is broad agreement between the 3 methods on cluster structure and allocations, with some differences particularly for allocation of individuals at the edges between larger clusters.

Fig 3 displays heatmaps of similarity matrices representing results from each of these algorithms, and also indicates the corresponding approximate posterior model probability, acting as a normalised weight $\hat{\mathcal{W}}_m$ for each model calculated from Eq 13 using the CH index. These normalised weights were: 0.35 for *k*-means; 0.30 for hierarchical clustering; and 0.35 for GMM. Taking an element-wise weighted average of these matrices, we calculated a consensus matrix *C* for which a heatmap is also presented in Fig 3. Finally, we applied SSMF to the consensus matrix *C* with $K_{BMA} = 5$ to generate a matrix $A^{BMA}$ of final cluster allocation probabilities.

Fig 4 presents the cluster allocations generated by *clusterBMA*, plotted according to each two-dimensional combination of the three retained principal components. In this plot the points

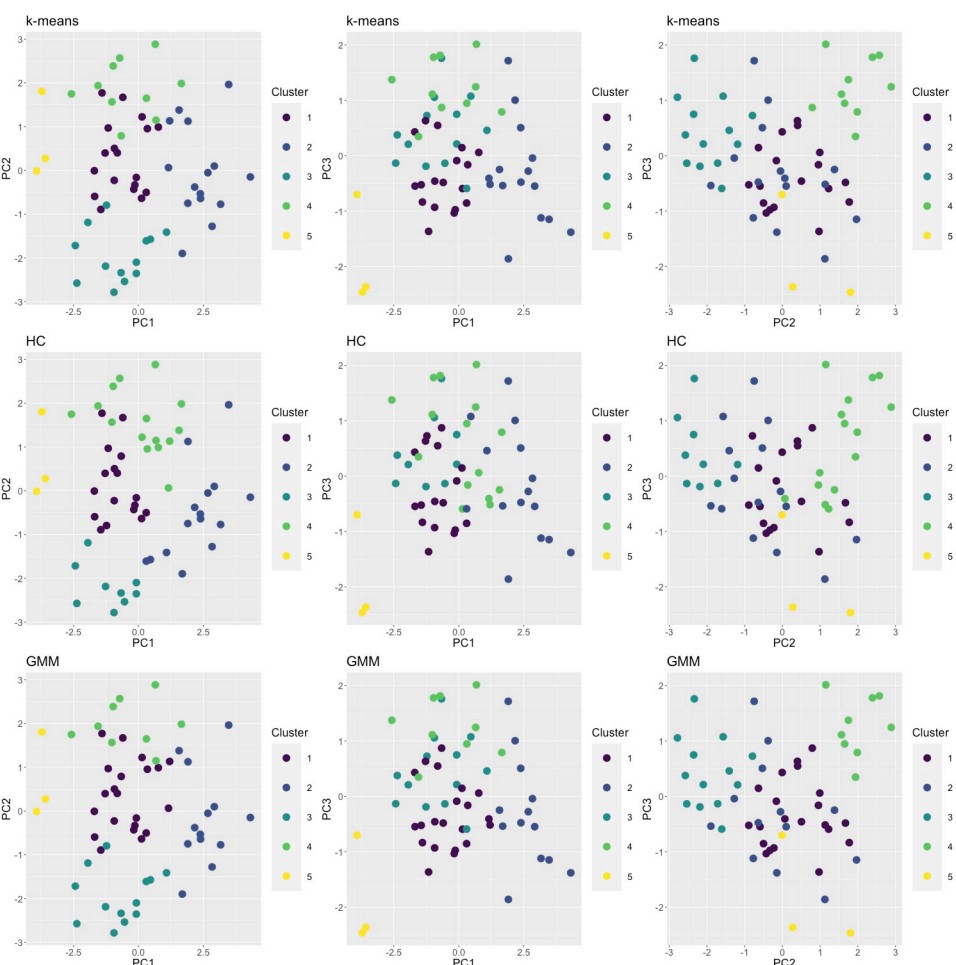

**Fig 2. 2D scatter plots of individuals by principal component scores, coloured by cluster membership, $K_m$ = 5.** Top row = $k$-means; middle row = HC; bottom row = GMM. Left column = PC1 v PC2; middle column = PC1 v PC3; right column = PC2 v PC3.

are scaled according to uncertainty of cluster allocation, $p(g_i \neq \hat{g}_i)$, with larger points representing higher uncertainty of allocation to a final cluster. These points with high uncertainty are largely at the boundaries between clusters, indicating the model-based uncertainty relating to clustering structure that would be ignored if choosing only one 'best' model out of the candidate algorithms. *clusterBMA* enables further analysis or prediction that can take this model-based uncertainty into account, which is not possible with other ensemble clustering methods. These outputs, including probabilistic allocation to averaged clusters and incorporation of model-based uncertainty, are useful for interpretation and statistical communication in the setting of applied health research and clinical practice. For instance, in scenarios where clusters might represent health phenotypes or clinical biomarkers, it is valuable for applied practitioners to understand the strength and uncertainty of allocations to clusters for the purpose of developing subsequent inferences and making assessments regarding clinical implications.

## 5. Discussion and conclusions

Clustering is a common goal for applied statistical analysis across many fields, and has grown in popularity alongside other unsupervised machine learning methods in recent years [62]. In

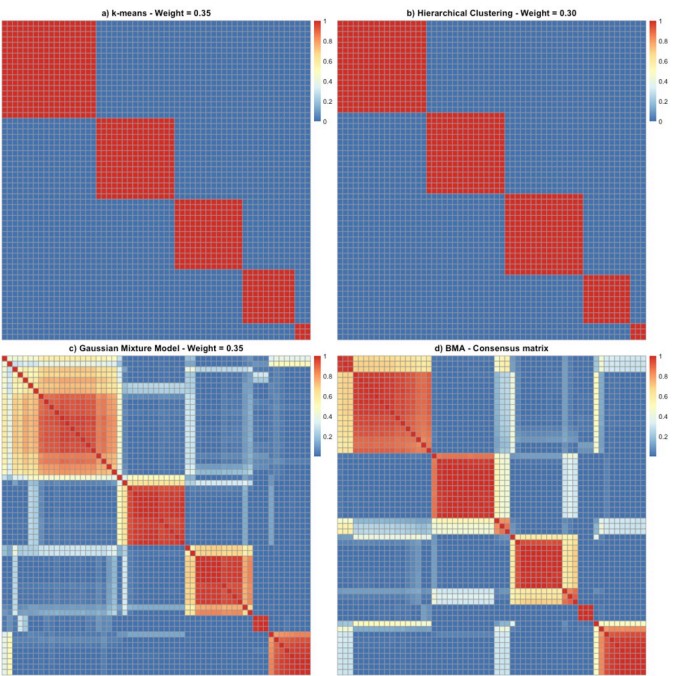

**Fig 3. Heatmaps of similarity matrices for each clustering algorithm and BMA consensus matrix.**

the context of health and medical research, clustering methods comprise a versatile set of statistical tools with a wide variety of potential applications including design of clinical trials [63], building data-driven profiles of individuals using functional biosignals data [64], and identifying clinical or epidemiological subtypes based on multivariate longitudinal observations [65]. Bayesian model averaging offers an intuitive and elegant framework to access more robust insights by combining inference across multiple clustering solutions.

Previous work has applied BMA within the model class of finite mixture models, weighting each model using an expression based on the Bayesian Information Criterion [8]. However, there has been limited development to date on methods to enable BMA across different classes of clustering models. We have introduced a novel Bayesian model averaging methodology, enabling a flexible approach for combining results from multiple unsupervised clustering methods which reduces the sensitivity of inferences to the analyst's choice of clustering

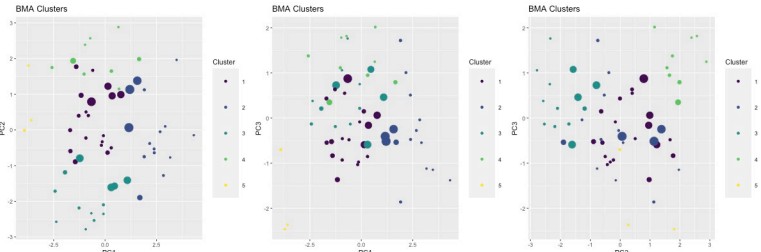

**Fig 4. 2D scatter plots of individuals by principal component scores, coloured by final cluster membership.** Point size is proportional to the uncertainty of cluster allocation, $p(\hat{g}_i \neq g_i)$. Larger points have greater uncertainty.

**Table 3. Cluster membership for different clustering algorithms & BMA clusters, $K_{BMA}$ = 5.**

| Cluster label | 1 | 2 | 3 | 4 | 5 |
|---:|---:|---:|---:|---:|---:|
| $k$-means | 18 | 15 | 13 | 10 | 3 |
| HC | 17 | 15 | 9 | 15 | 3 |
| GMM | 21 | 13 | 13 | 9 | 3 |
| BMA | 19 | 15 | 13 | 9 | 3 |

algorithm. We have extended on previous work to approximate the posterior model probability for each model using a normalised weight based on cluster internal validation indices. This approach allows BMA to be implemented across results from different clustering methods. A consensus matrix is calculated as an element-wise weighted average of the similarity matrices from each input algorithm. Final probabilistic cluster allocations are generated by applying symmetric simplex matrix factorisation to this consensus matrix.

Our principal simulation study has shown that relative to other available methods for ensemble clustering, *clusterBMA* offers equal or better performance among different conditions for simulated data, and consistently outperforms other ensemble clustering methods for high-dimensional data with low cluster separation, which is reflective of data features that are common for many real world clustering scenarios (Table 3). In addition to this strong benchmarking performance, our method implements a number of attractive features that are not available in competing methods, including weighted averaging across models, generation of probabilistic inferences, and quantification of model-based uncertainty. Our case study and simulation studies have demonstrated the capacity of *clusterBMA* to combine clustering solutions from different clustering algorithms and with different numbers of clusters. These applications demonstrate identification of cluster allocations with higher model-based uncertainty that are typically concentrated at the boundaries between clusters, where there tends to be a higher level of disagreement between multiple clustering solutions. Our method captures this uncertainty relating to clustering structure that would be ignored when using results from one 'best' algorithm, or when using a consensus clustering method that does not incorporate model-based uncertainty. This method has flexibility to accommodate different numbers of clusters in each candidate model, and does not require cluster labels to be aligned across models.

As in most statistical and machine learning methods, many elements of clustering analysis require the analyst to make reasoned and considered choices including the choice of algorithms, validation indices, and numbers of clusters. Our approach makes these aspects of the clustering process more transparent, which would otherwise tend to be hidden from the presentation of analysis and results. The outputs from *clusterBMA* highlight variation in clustering results across different algorithms, assessment of the quality of the contribution from each algorithm, and combination of these results in a principled way that allows subsequent inferences to calibrate for the uncertainty that arises in an "M-open" candidate model space. When making modelling decisions, the analyst's due diligence should include considered choice of a CIVI for weighting each clustering model according to traits which reflect the clustering objectives of the analysis. Clustering algorithms and CIVIs have different use cases which should be selected to align with the objectives of a given analysis. With this method, as with all Bayesian model averaging, the principle of 'Garbage In, Garbage Out' applies and the onus remains on the analyst to only average across input models that seem plausible and each provide useful insight into the data. Including poor models as an input could dilute the quality of the model averaging results.

For approximating posterior model probability, the two measures we have recommended (the CH and S_Dbw indices) have demonstrated good performance in a range of settings [25, 27], but there are a range of alternative CIVIs that could be tested and considered in this setting. While these two indices are likely to be useful for weighting each model in a wide range of scenarios for *clusterBMA*, there are some caveats to their use. For instance, the CH index as typically applied using Euclidean distance will tend to be biased towards solutions with more spherical clusters [26]. This is likely to be a reasonable assumption for many clustering applications, however in situations where strongly non-spherical clusters are suspected, a different CIVI should be used that accommodates this analytic objective. The S_Dbw index can have a very high computational cost with large datasets [66], and may face computational obstacles with density calculations for sparse, high dimensional data. A range of other CIVIs are available to use as weighting measures in *clusterBMA*, which are offered through the R package *clusterCrit* which offers accompanying documentation on the characteristics of each index and whether it is to be maximised or minimised [28].

In this setting we have assigned equal prior probability to each input model; however, alternative approaches for assigning priors could be considered. For instance, priors could be selected to penalise for the number of parameters in each model in order to preference model parsimony, or a vector of prior weights could be manually provided to assign greater weight to selected input models in the scenario that one particular clustering structure is known to be more useful for a particular dataset.

A limitation of this method is that uncertainty quantification is implemented as a point estimate based on the probability that the true cluster allocation is not equal to the estimated cluster allocation, $p(g_i \neq \hat{g}_i)$. This approach has been used elsewhere [30], and incorporates both the uncertainty of allocation from probabilistic clustering inputs from "soft" clustering algorithms, as well as the uncertainty arising from ambiguity across multiple clustering models. However, it does not fully characterise the probability distributions corresponding to probabilistic cluster allocations and instead only a point estimate is available to measure the degree of this uncertainty.

For all of the applications presented in this paper, computing times for *clusterBMA* are typically of the order of seconds, rather than minutes or hours. The most computationally expensive part of the *clusterBMA* pipeline is symmetric simplex matrix factorisation, where gradient descents in each iteration of expectation maximisation (EM) have computational complexity $O(n^2 d)$ [29]. In addition to the sample size $n$ and dimensionality $d$, the computation time will also be dependent on the number of EM iterations—by default this is set to 5000 in the R package, but this is likely to be higher than necessary for many use cases, and can be adjusted by the user as needed. Another aspect of computational complexity here is that when the sample size is very large, this can make the similarity matrix computationally prohibitive. An alternative approach that has been proposed for such scenarios is using random feature maps [29, 67]. We have found that computation times are short using a personal computer for most use cases, though applications with very large datasets may require adjustments as discussed above, or implementation using high performance computing platforms.

While in the current work we have compared *clusterBMA*'s performance against four ensemble clustering methods implemented in the *diceR* package, there are many other ensemble clustering methods against which our method could be compared [4]. Additionally, other metrics than the Adjusted Rand Index could be considered to compare different aspects of relative performance between *clusterBMA* and other ensemble clustering methods. However, overall we have demonstrated that *clusterBMA* performs well across a variety of simulated data scenarios relative to other methods, and to our knowledge the unique benefits and features of our method described in this paper are not available in any other ensemble clustering methods.

This framework could be extended in future to be more 'fully Bayesian', accommodating results from Bayesian clustering algorithms as inputs, and enabling more complete characterisation of uncertainty in probability distributions for cluster allocation probabilities. As one approach, this could be accomplished by using the existing pipeline with draws from Markov Chain Monte Carlo sampling for results of Bayesian clustering models. This could potentially accommodate inputs from both frequentist and Bayesian clustering algorithms as inputs, by matching MCMC samples with an appropriate number of replicates of results from frequentist clustering algorithms. This extension could enable more complete characterisation of the model-based uncertainty relating to the probability distributions for probabilities of final allocations. Another avenue that could be explored for approximating the marginal likelihood for clustering models could consider the equivalence between the marginal likelihood and exhaustive leave-$p$-out cross validation, investigating the validity of this approach in the clustering setting and for methods without likelihood terms [68]. Future work could also explore the performance and utility of different CIVIs in clustering scenarios with different data characteristics and analytic objectives.

This method is implemented in an accompanying R package, *clusterBMA* [31]. It offers an intuitive, flexible and practical framework for analysts to combine inferences across multiple clustering algorithms with quantified model-based uncertainty. Future development in this space could enable additional functionality such as accommodating sampling-based input from Bayesian clustering algorithms, incorporating informative prior information, and exploring the utility of alternative internal validation measures for the approximation of posterior model probability.

## Supporting information

**S1 File.**
(PDF)

## Acknowledgments

We are grateful to the editor Dariusz Siudak, and reviewers Jonathan M. Keith and Virgilio Gómez-Rubio for their thoughtful feedback in supporting revisions and improvements to this manuscript.

We extend our gratitude to the Longitudinal Adolescent Brain Study (LABS) participants and their caregivers.

## Author Contributions

**Conceptualization:** Owen Forbes, Edgar Santos-Fernandez, Paul Pao-Yen Wu, Kerrie Mengersen.

**Data curation:** Owen Forbes, Paul E. Schwenn, Jim Lagopoulos, Lia Mills, Dashiell D. Sacks, Daniel F. Hermens.

**Formal analysis:** Owen Forbes, Paul Pao-Yen Wu, Kerrie Mengersen.

**Funding acquisition:** Owen Forbes.

**Investigation:** Owen Forbes, Hong-Bo Xie, Kerrie Mengersen.

**Methodology:** Owen Forbes, Edgar Santos-Fernandez, Paul Pao-Yen Wu, Hong-Bo Xie, Paul E. Schwenn, Kerrie Mengersen.

**Project administration:** Owen Forbes, Paul E. Schwenn, Jim Lagopoulos, Lia Mills, Dashiell D. Sacks, Daniel F. Hermens, Kerrie Mengersen.

**Resources:** Kerrie Mengersen.

**Software:** Owen Forbes, Edgar Santos-Fernandez.

**Supervision:** Edgar Santos-Fernandez, Paul Pao-Yen Wu, Hong-Bo Xie, Paul E. Schwenn, Kerrie Mengersen.

**Validation:** Owen Forbes, Kerrie Mengersen.

**Visualization:** Owen Forbes.

**Writing – original draft:** Owen Forbes.

**Writing – review & editing:** Owen Forbes, Edgar Santos-Fernandez, Paul Pao-Yen Wu, Hong-Bo Xie, Paul E. Schwenn, Jim Lagopoulos, Lia Mills, Dashiell D. Sacks, Daniel F. Hermens, Kerrie Mengersen.

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
