## [Decision Letter · Decision Letter 0]

22 May 2023

PONE-D-23-08970clusterBMA: Bayesian model averaging for clusteringPLOS ONE

Dear Dr. Forbes,

Thank you for submitting your manuscript to PLOS ONE. After careful consideration, we feel that it has merit but does not fully meet PLOS ONE’s publication criteria as it currently stands. Therefore, we invite you to submit a revised version of the manuscript that addresses the points raised during the review process.

Please note that the review of Reviewer #2 is in the attached file. Please also pay attention to the two comments of the Editor.

We look forward to receiving your revised manuscript.

Kind regards,

Dariusz Siudak, Ph.D., DSc.

Academic Editor

PLOS ONE

Journal Requirements:

Additional Editor Comments (if provided):

In addition to the comments of both reviewers, please consider my two additional comments:

1. The aim of the study should be stated in the introduction.

2. Section 5 (Discussion) should either be divided into two parts Discussion and Conclusions, or Section 5 should be referred to as Discussion and Conclusions.

Reviewers' comments:

Reviewer's Responses to Questions

**Comments to the Author**

1. Is the manuscript technically sound, and do the data support the conclusions?

Reviewer #1: Yes

Reviewer #2: Yes

2. Has the statistical analysis been performed appropriately and rigorously? 

Reviewer #1: Yes

Reviewer #2: Yes

3. Have the authors made all data underlying the findings in their manuscript fully available?

Reviewer #1: No

Reviewer #2: Yes

4. Is the manuscript presented in an intelligible fashion and written in standard English?

Reviewer #1: Yes

Reviewer #2: Yes

5. Review Comments to the Author

Reviewer #1: The paper describes a new technique for ensemble clustering, using a weighted sum of similarity matrices. The proposed method is almost naive in its simplicity, but the authors thoroughly and convincingly demonstrate that their proposed method outperforms several existing ensemble clustering methods. It's a clear and well written paper that covers all bases, and consequently I can suggest only trivial corrections, as follows.

1) L53: "used a" should be "used as a"

2) L117: The words "measure" and "metric" are used throughout the paper, interchangeably and without their mathematical meanings, with the first use occurring on L117. I'd prefer to see these words used only in their mathematical sense in a statistical paper, with an alternative word used in this context. For example, the word "property" would be suitable on L117.

3) L135-6: "member cluster" should be "member of cluster".

4) L195-6: The word "goal" is repeated.

5) L251: The word "and" appears twice, as though an item has been added to the end of the list. The second "and" should be preceded by a semi-colon, not a comma, or else commas should be used in both places.

6) L270: "Eq. 12" should be "Eq. 13".

7) L296-7: Perhaps the number of clusters used for the BMA clustering should be larger than any of the numbers of clusters used in the input models, because it could happen that one method succeeds in resolving one cluster into sub-clusters, whereas another method resolves a different cluster into sub-clusters. It could happen that different clustering methods are better at resolving different types of clusters.

8) L463: A left parenthesis appears to be missing.

9) L510: "clusers" should be "clusters".

10) L546: Delete "the" before "Fig".

11) L578-9: Two periods should be commas.

12) L671: "variety simulated" should be "variety of simulated".

13) Figure quality is low and text in figures is difficult to read - I presume higher quality figures will be provided.

Reviewer #2: The authors have presented a new method for classification based on Bayesian model averaging. After reading the paper, I think that it still requires substantial work before it can eventually be published. Please, find some detailed comments in the attached paper..

6. PLOS authors have the option to publish the peer review history of their article (what does this mean?). If published, this will include your full peer review and any attached files.

Reviewer #1: **Yes: **Jonathan M. Keith

Reviewer #2: **Yes: **Virgilio Gómez-Rubio

---

## [Author Response · Author response to Decision Letter 0]

26 May 2023

Response to Reviewers comments

Title: clusterBMA: Bayesian model averaging for clustering

Thank you for reviewing our paper. We appreciate the valuable feedback and suggestions. I have attached the revised paper and modified figures, and have outlined below point by point how the paper has been modified to address the comments of the editor, and Reviewers #1 and #2.

Editor’s Comments

Editor: 1. The aim of the study should be stated in the introduction.

Thank you for your recommendation. We have added the following sentence to the introduction to clarify the aim of the study at L49: 

“The aim of this work is to propose a BMA framework that can effectively combine results across multiple unsupervised clustering algorithms. We showcase the performance and effectiveness of this framework through various clustering applications, including simulated data experiments and a real-world case study involving neuroscientific data.”

Editor: 2. Section 5 (Discussion) should either be divided into two parts Discussion and Conclusions, or Section 5 should be referred to as Discussion and Conclusions.

We have changed the name of Section 5 to “Discussion and Conclusions”.

 

Reviewer #1 Comments

Reviewer #1: The paper describes a new technique for ensemble clustering, using a weighted sum of similarity matrices. The proposed method is almost naive in its simplicity, but the authors thoroughly and convincingly demonstrate that their proposed method outperforms several existing ensemble clustering methods. It's a clear and well written paper that covers all bases, and consequently I can suggest only trivial corrections, as follows.

Thank you for your feedback. We are pleased to hear that you found our ensemble clustering approach satisfactory.

Reviewer #1: 1) L53: "used a" should be "used as a"

Thank you for spotting this issue. We have fixed it in the manuscript.

Reviewer #1: 2) L117: The words "measure" and "metric" are used throughout the paper, interchangeably and without their mathematical meanings, with the first use occurring on L117. I'd prefer to see these words used only in their mathematical sense in a statistical paper, with an alternative word used in this context. For example, the word "property" would be suitable on L117.

Thank you for your comment. As suggested, we have now used the term “property” or “quantity” when referring to \\Delta in the context of BMA. 

Consistent with other literature on internal validation measures, we have revised the terminology to use either “measure” or “index” (avoiding “metric”) when referring to cluster internal validation indices (see e.g., Liu, Y., Li, Z., Xiong, H., Gao, X., & Wu, J. (2010, December). Understanding of internal clustering validation measures. In 2010 IEEE international conference on data mining (pp. 911-916). IEEE.)

Reviewer #1: 3) L135-6: "member cluster" should be "member of cluster".

Fixed, thank you.

Reviewer #1: 4) L195-6: The word "goal" is repeated.

We have fixed that in the manuscript. Thank you.

Reviewer #1: 5) L251: The word "and" appears twice, as though an item has been added to the end of the list. The second "and" should be preceded by a semi-colon, not a comma, or else commas should be used in both places.

Fixed, thank you.

Reviewer #1: 6) L270: "Eq. 12" should be "Eq. 13".

We have fixed that in the manuscript. Thank you.

Reviewer #1: 7) L296-7: Perhaps the number of clusters used for the BMA clustering should be larger than any of the numbers of clusters used in the input models, because it could happen that one method succeeds in resolving one cluster into sub-clusters, whereas another method resolves a different cluster into sub-clusters. It could happen that different clustering methods are better at resolving different types of clusters.

Thank you for your comment. We have added the following sentence at L313 to reflect this point: 

“Given the reduction of redundant clusters with L2 regularisation, another possible heuristic for choosing $K_{BMA}$ would be to choose a larger number of clusters than the largest $K_m$ across the input models, accommodating the possibility of different sets of sub-clusters appearing across different input models.”

Reviewer #1: 8) L463: A left parenthesis appears to be missing.

Fixed, thank you.

Reviewer #1: 9) L510: "clusers" should be "clusters".

Fixed, thank you.

Reviewer #1: 10) L546: Delete "the" before "Fig".

Fixed, thank you.

Reviewer #1: 11) L578-9: Two periods should be commas.

Fixed, thank you.

Reviewer #1: 12) L671: "variety simulated" should be "variety of simulated".

Fixed, thank you.

Reviewer #1: 13) Figure quality is low and text in figures is difficult to read - I presume higher quality figures will be provided.

Yes, higher quality figures will be included in the final publication.

 

Reviewer #2 Comments

Reviewer #2: Section 2.1. The method overview is fine but it comes too early in the paper, which makes it difficult to understand. Also, using A^m and S^m are a bit confusing and I would change it to A_m (as in K_m and other quantities that appear in the paper). Perhaps this section could be moved somewhere else in the paper.

Thank you for your helpful comments. 

The intention of presenting the methods overview here at the start of the methods is to provide a high-level scaffold of how the methods fit together, so that the reader has an overall schema in which to understand the detail of the methods subsequently described. To aid the understanding of the overview in this context, we have added the following sentences at L96:

“Here we present a high level overview of the methodological steps involved in clusterBMA. The intention is to provide a road map for the reader, making the detailed explanations of each individual step in the following sections easier to understand in the broader context of this framework.”

We have also adjusted the formatting of Table 1 so that it no longer breaks up the methods overview bullet points, to enable improved clarity. 

Using the superscripts A^m and S^m to index the models m = 1, … , M follows convention for matrix notation in the clustering literature, and indexing by model in the superscript allows for indexing by matrix elements in the subscript e.g. S^m_{ij} representing the probability that points i and j are allocated together in model m: 

While we appreciate the suggestion, we have chosen to maintain the notation in line with the traditional convention in the clustering literature. However, we are open to making changes if the editor believes it would enhance clarity.

Reviewer #2: Section 2.2. The first sentence is a bit cryptic. I would include a description of what \\Delta actually represents and provide some context.

Thank you for your comment. To improve clarity in this section, we have amended the first sentence at L123:

“Consider a quantity of interest \\Delta which is present in every model across a set of candidate models for a given analysis.”

Reviewer #2: Section 2.2, eq (1). I believe that the left hand side should be p(\\Delta = \\Delta_m \\par Y) as the right hand side involves \\Delta_m.

Thank you for your comment. In this case, the term on the RHS \\sum_{m=1}^M sums over the posterior estimates for \\Delta_m in each model, indicating that left hand side p(\\Delta | Y) represents the posterior estimate for delta averaged across all models, rather than p(\\Delta = \\Delta_m | Y) as suggested.

Reviewer #2: Section 2.2, eq. (2). are the probabilities in the right hand side correct? Why conditioning on i and j (in addition to M_m)? Also, I would expect the probability of two observations being in the same cluster to be computed using the joint posterior probabilities and not the marginal ones.

Thank you for your comment. I may be mistaken, but I think you are referring to Eq. 3 (not Eq. 2) featuring i and j in the RHS:

Here the terms on the RHS conditioned on i and j are defined as follows: p(g_k|i, M_m) indicates the probability that point i is a member of cluster g_k in model M_m, and p(g_k|j, M_m) indicates the probability that point j is a member of cluster g_k in model M_m. 

To add clarity, we have expanded the following sentence at L142:

“…where $g_k$ is the $k$th cluster, and $p(g_k|i, \\mathcal{M}_m)$ indicates the probability that point $i$ is a member of cluster $g_k$ in model $\\mathcal{M}_m$.”

This method for calculating the probability of two observations being in the same cluster follows the implementation in the following two references, which we have included citations for in this paragraph. 

Russell, N., Murphy, T. B., & Raftery, A. E. (2015). Bayesian model averaging in model-based clustering and density estimation. arXiv preprint arXiv:1506.09035.

Fern, X. Z., & Brodley, C. E. (2003). Random projection for high dimensional data clustering: A cluster ensemble approach. In Proceedings of the 20th international conference on machine learning (ICML-03) (pp. 186-193).

Reviewer #2: Section 2.3, eq. (5). Although I can see the reason to use this approximation. How do the estimates of the effective number of parameters affect the estimates of P(M_m |Y)?

Thank you for your comment. To add clarity, we have added this sentence at L162:

“This is the negative of the usual construction of the BIC, and a larger number of model parameters $\\kappa_m$ will result in a smaller estimate for the approximated posterior model probability of model $M_m$.”

Reviewer #2: Section 2.3, eq. (13). I see that that authors want (re-scaled) W_m to play the role of the marginal likelihood in their weighting scheme. However, is really the right hand side an approximation to the marginal likelihood? Is there a paper in which this approximation is actually developed and discussed?

Thank you for your comment. Treating the normalised / re-scaled weight W_m as an approximation of the marginal likelihood is also applied in Russell et al. (2015). – see Eq. 5, where W_m is calculated as : 

As noted in the paragraph starting at L175:

“While ideally we would like to use a measure such as the BIC with strong theoretical support for approximating the marginal likelihood to weight each model, the BIC is not viable for our application of weighting solutions generated from multiple classes of clustering algorithm.”

To clarify that Eq. 13 represents our proposed approximation for the marginal likelihood based on a weighting term using clustering internal validation indices (which we are developing and discussing in this paper) we have amended the following sentence at L276:

“Having chosen a CIVI to act as a weighting variable $\\mathcal{W}_m$ for each model, we propose the following normalised weight $\\hat{\\mathcal{W}}_m$ as an approximation for the marginal likelihood $P(Y \\lvert \\mathcal{M}_m)$ for each model:”

Reviewer #2: Sections 3.3 and 3.4. Are there tables similar to Table 2 for these scenarios? I have not found them in the Supplementary materials.

No, equivalents to Table 2 are not presented for Simulation Studies 2 or 3, as these studies were not designed to compare the performance of clusterBMA with other ensemble methods. 

We have expanded the following paragraph in the discussion at L697 to address that future work could consider a broader set of comparisons for clusterBMA against other ensemble clustering methods:

“While in the current work we have compared \\textit{clusterBMA}'s performance against four ensemble clustering methods implemented in the \\textit{diceR} package, there are many other ensemble clustering methods against which our method could be compared \\citep{golalipour2021clustering}. Additionally, other metrics than the Adjusted Rand Index could be considered to compare different aspects of relative performance between \\textit{clusterBMA} and other ensemble clustering methods. However, overall simulation study 1 demonstrated that \\textit{clusterBMA} performs well across a variety of simulated data scenarios relative to other methods, and to our knowledge the unique benefits and features of our method described in this paper are not available in any other ensemble clustering methods.”

Reviewer #2: Section 4. What is the number of features used in the analysis? In Section 4.2, it says “… 8 summary features in the frequency domain.” but then a dimension reduction is used. But the resulting dimension of the feature space is not mentioned at all.

Thank you for your comment. At L532 we have added the following sentence:

“From the principal component analysis, the first three principal components were retained which together explained 80.6\\% of the overall variance.”

Also as noted at L543 “Fig 2 presents the clustering results from each algorithm, plotted according to each two-dimensional combination of the three retained principal components.”

Reviewer #2: Section 4. How fast is this method compared to others? Table 3 shows that GMM provides a very similar clustering as BMA (in terms of subjects per group).

Thank you for your comment. While GMM produces similar numbers of individuals in each cluster, Figures 2 and 4 demonstrate that the BMA solution and GMM partition have differences in cluster allocations, and the GMM partition does not include model-based uncertainty which is captured in the BMA solution. 

We have added the following paragraph to the Discussion at L682 to address computation time/complexity. 

“For all of the applications presented in this paper, computing times for clusterBMA are typically of the order of seconds, rather than minutes or hours. The most computationally expensive part of the clusterBMA pipeline is symmetric simplex matrix factorisation, where gradient descents in each iteration of expectation maximisation (EM) have computational complexity $O(n^2d)$ \\citep{duan2020latent}. In addition to the sample size $n$ and dimensionality $d$, the computation time will also be dependent on the number of EM iterations – by default this is set to 5000 in the R package, but this is likely to be higher than necessary for many use cases, and can be adjusted by the user as needed. Another aspect of computational complexity here is that when the sample size is very large, this can make the similarity matrix computationally prohibitive. An alternative approach that has been proposed for such scenarios is using random feature maps \\citep{duan2020latent, rahimi2007random}. We have found that computation times are short using a personal computer for most applications, though applications with very large datasets may require adjustments as discussed above, or implementation using high performance computing platforms.”

Reviewer #2: Figures 2 and 4 use colors for clusters 1 and 5 that make them difficult to identify.

Thank you for your comment. We have amended these figures to use the “viridis” color palette, which is designed to have clear visual differences and be easier to read by those with colorblindness: https://cran.r-project.org/web/packages/viridis/vignettes/intro-to-viridis.html

Reviewer #2: Also, the authors present a real example that can be tackled with standard methods. I would choose a different example in which their method ‘shines’ over existing methods.

The use of this specific case study based on EEG data in adolescents is partly motivated by forming part of a suite of work contributing to the lead author’s PhD thesis by publication, focused on neuroscientific data in young people.

While existing methods could plausibly be used for this case study, alternative methods do not offer probabilistic allocation to combined clusters or quantification of model-based uncertainty, both of which are useful features for possible clinical applications, as addressed at L574: 

“These outputs, including probabilistic allocation to averaged clusters and incorporation of model-based uncertainty, are useful for interpretation and statistical communication in the setting of applied health research and clinical practice. For instance, in scenarios where clusters might represent health phenotypes or clinical biomarkers, it is valuable for applied practitioners to understand the strength and uncertainty of allocations to clusters for the purpose of developing subsequent inferences and making assessments regarding clinical implications.”

Simulation Study 1 has been designed to compare the performance of clusterBMA with existing ensemble methods, demonstrating that it ‘shines’ compared to other methods particularly for high dimensional data with low separation between clusters.

Reviewer #2: Bibliography: Russell et al. Has the wrong arXiv code as a dot is missing.

Thank you for your comment. We have corrected this reference.

---

## [Decision Letter · Decision Letter 1]

13 Jun 2023

PONE-D-23-08970R1clusterBMA: Bayesian model averaging for clusteringPLOS ONE

Dear Dr. Forbes,

Thank you for submitting your manuscript to PLOS ONE. After careful consideration, we feel that it has merit but does not fully meet PLOS ONE’s publication criteria as it currently stands. Therefore, we invite you to submit a revised version of the manuscript that addresses the points raised during the review process.

I agree with reviewer #2's comment regarding Section 2.1. Since many symbols from Section 2.1 are introduced in subsequent sections, I propose moving Section 2.1 after Section 2.4.

We look forward to receiving your revised manuscript.

Kind regards,

Dariusz Siudak, Ph.D., DSc.

Academic Editor

PLOS ONE

Journal Requirements:

Reviewers' comments:

Reviewer's Responses to Questions

**Comments to the Author**

1. If the authors have adequately addressed your comments raised in a previous round of review and you feel that this manuscript is now acceptable for publication, you may indicate that here to bypass the “Comments to the Author” section, enter your conflict of interest statement in the “Confidential to Editor” section, and submit your "Accept" recommendation.

Reviewer #1: All comments have been addressed

Reviewer #2: (No Response)

2. Is the manuscript technically sound, and do the data support the conclusions?

Reviewer #1: Yes

Reviewer #2: Yes

3. Has the statistical analysis been performed appropriately and rigorously? 

Reviewer #1: Yes

Reviewer #2: Yes

4. Have the authors made all data underlying the findings in their manuscript fully available?

Reviewer #1: No

Reviewer #2: (No Response)

5. Is the manuscript presented in an intelligible fashion and written in standard English?

Reviewer #1: Yes

Reviewer #2: Yes

6. Review Comments to the Author

Reviewer #1: (No Response)

Reviewer #2: I still think that Section 2.1 should be moved from there. As it is now, the only option for the reader is to skip it on a first read. Hence, there is no point in keeping it where it is. However, I'll leave that decision to the editor.

7. PLOS authors have the option to publish the peer review history of their article (what does this mean?). If published, this will include your full peer review and any attached files.

Reviewer #1: **Yes: **Jonathan M. Keith

Reviewer #2: **Yes: **Virgilio Gómez-Rubio

---

## [Author Response · Author response to Decision Letter 1]

15 Jun 2023

Response to Reviewers comments

Title: clusterBMA: Bayesian model averaging for clustering

Thank you for reviewing our paper. I have attached the revised paper and have outlined below how the paper has been modified to address the comments of the editor, and Reviewer #2.

Editor’s Comments

Editor: I agree with reviewer #2's comment regarding Section 2.1. Since many symbols from Section 2.1 are introduced in subsequent sections, I propose moving Section 2.1 after Section 2.4.

Reviewer #2: I still think that Section 2.1 should be moved from there. As it is now, the only option for the reader is to skip it on a first read. Hence, there is no point in keeping it where it is. However, I'll leave that decision to the editor.

Thank you both for your comments. We have made the suggested change in the attached manuscript, and moved the overview to section 2.4.

Kind Regards,

Owen Forbes (on behalf of the co-authors)

---

## [Editor Report · Decision Letter 2]

19 Jun 2023

clusterBMA: Bayesian model averaging for clustering

PONE-D-23-08970R2

Dear Dr. Forbes,

We’re pleased to inform you that your manuscript has been judged scientifically suitable for publication and will be formally accepted for publication once it meets all outstanding technical requirements.

Kind regards,

Dariusz Siudak, Ph.D., DSc.

Academic Editor

PLOS ONE
---

## [Editor Report · Acceptance letter]

11 Aug 2023

PONE-D-23-08970R2 

clusterBMA: Bayesian model averaging for clustering 

Dear Dr. Forbes:

I'm pleased to inform you that your manuscript has been deemed suitable for publication in PLOS ONE. Congratulations! Your manuscript is now with our production department. 

Kind regards, 

on behalf of

Dr. Dariusz Siudak 

Academic Editor

PLOS ONE